# Diagnosing Onychomycosis: What’s New?

**DOI:** 10.3390/jof8050464

**Published:** 2022-04-29

**Authors:** Aditya K. Gupta, Deanna C. Hall, Elizabeth A. Cooper, Mahmoud A. Ghannoum

**Affiliations:** 1Department of Medicine, Division of Dermatology, University of Toronto School of Medicine, Toronto, ON M5S 3H2, Canada; 2Mediprobe Research Inc., London, ON N5X 2P1, Canada; dhall@mediproberesearch.com (D.C.H.); lcooper@mediproberesearch.com (E.A.C.); 3Center for Medical Mycology, Department of Dermatology, Case Western Reserve University, Cleveland, OH 44106, USA; mag3@case.edu; 4Department of Dermatology, University Hospitals Cleveland Medical Center, Cleveland, OH 44106, USA

**Keywords:** onychomycosis, diagnosis, polymerase chain reaction, spectroscopy, artificial intelligence, microscopy, tomography

## Abstract

An overview of the long-established methods of diagnosing onychomycosis (potassium hydroxide testing, fungal culture, and histopathological examination) is provided followed by an outline of other diagnostic methods currently in use or under development. These methods generally use one of two diagnostic techniques: visual identification of infection (fungal elements or onychomycosis signs) or organism identification (typing of fungal genus/species). Visual diagnosis (dermoscopy, optical coherence tomography, confocal microscopy, UV fluorescence excitation) provides clinical evidence of infection, but may be limited by lack of organism information when treatment decisions are needed. The organism identification methods (lateral flow techniques, polymerase chain reaction, MALDI-TOF mass spectroscopy and Raman spectroscopy) seek to provide faster and more reliable identification than standard fungal culture methods. Additionally, artificial intelligence methods are being applied to assist with visual identification, with good success. Despite being considered the ‘gold standard’ for diagnosis, clinicians are generally well aware that the established methods have many limitations for diagnosis. The new techniques seek to augment established methods, but also have advantages and disadvantages relative to their diagnostic use. It remains to be seen which of the newer methods will become more widely used for diagnosis of onychomycosis. Clinicians need to be aware of the limitations of diagnostic utility calculations as well, and look beyond the numbers to assess which techniques will provide the best options for patient assessment and management.

## 1. Introduction

Onychomycosis, or tinea unguium, is one of the most prevalent nail pathologies across the globe, with significant impact on quality of life [1]. The dermatophytes *Trichophyton rubrum* and *T. interdigitale* are the most common organisms causing infection, but other dermatophytes, non-dermatophyte molds (NDMs) and yeasts may also cause infection [2]. Mixed infections produce clinical difficulty, as not all organisms respond equally to treatment. The ideal diagnosis should be quick and thorough, including the identification of all active infecting species to allow for selection of the optimal antifungal treatment.

Emerging diagnostic technologies divide into visual identification methods that increase the visibility of nail dystrophy and/or fungal structures (e.g., hyphae/yeast) indicating onychomycosis presence, and organism identification methods which provide the genus/species of infecting fungi (Table 1). Use of these methods, and their limitations, must be well-understood by clinicians for optimum diagnosis and treatment of onychomycosis. These newer methodologies may become more prevalent as research continues, and it will be clinician use and experience that will determine the future mainstream diagnostic options for onychomycosis.

## 2. Established Methods: KOH, Histopathology, Culture

**Potassium hydroxide (KOH)** testing uses a KOH solution (typically 10–20% KOH) to reduce keratin in nail samples, improving visualization of fungal structures under light microscopy (Figure 1A). A variety of stains may be added with KOH to increase visualization. Calcofluor white (CW) highlights hyphal walls from surrounding debris under a fluorescent microscope (Figure 1B). KOH is both quick (approximately 30 min [3,4]) and inexpensive [3,5]. The procedure can be carried out in an office or laboratory. The accuracy depends on proper collection of samples, as well as the examiner’s experience [6]. Despite the best collection, typically only a portion of a sample is used for examination, with the remainder reserved for culture or other testing, and, by chance, the examined portion may not contain fungal structures. 

**Histopathological examination** can be performed on intact nail plate portions or nail biopsy specimens, using periodic acid-Schiff (PAS) or Grocott methenamine silver (GMS) to stain fungal elements for increased visibility under microscopic exam (Figure 1C). Specimens require significant laboratory preparation, including embedding in media prior to sectioning for examination [8]. Fungal elements can be visualized within the nail unit, indicating an active infection, or on nail margins, suggesting a contaminant. Nail dystrophy associated with infection may also be visible. Histopathology is reportedly the most sensitive of the conventional onychomycosis diagnostic methods; however, as with KOH, it does not provide the identity or viability of the organism [9,10]. Submission of a nail plate alone may not detect nail bed infection.

**Fungal culture** attempts to grow viable fungi from a nail sample using culture media. A wide variety of culture media exist for growing fungi and may be augmented with antibiotics to provide a selective growth environment (e.g., Sabouraud dextrose agar with cycloheximide, chloramphenicol and gentamycin for deterrence of bacteria and most non-dermatophyte filamentous fungi). The growth process can require up to a month or longer to establish the final result [3,4,5,11,12]. Examination of a culture and its microscopic fungal morphology can allow identification of pathogen subtype in most cases when observed by an experienced technician [3,13,14]. Identified fungi, though viable, may be a contaminant rather than a causative agent; technician experience and physician experience must combine to assess likelihood of the fungal agent as the cause of infection.

## 3. Established Methods as ‘Gold Standard’?

The conventional macroscopic and microscopic methods described above have been considered ‘gold standard’ methods for determining whether an individual has a nail fungal infection or not. The ‘gold standard’ or ‘reference standard’ title is ideally reserved for diagnostic methods which have high sensitivity for detecting onychomycosis and high specificity in ruling out onychomycosis where it is not present. Unfortunately, these methods fall short of being ideal but have been considered as ‘standard’ due to the lack of available alternative methods with demonstrated efficacy. Culture is known for having a high rate of false negative findings and would need to be combined with KOH and/or PAS to merit any ‘reference standard’ designation. Even so, the combination of a positive KOH or PAS with negative culture, or with a suspected contaminant culture result, gives very little information when clinicians seek to provide treatment. 

Calculations of sensitivity and specificity to assess diagnostic utility depend upon having a reliable reference standard of infection status, but the above limitations provide a significant obstacle to using these methods as reference standards. KOH sensitivity has been calculated from as low as 33.7% [15] and up to 93%, [9] with specificity from 38% [9] to 100% [16,17,18]. Through meta-analysis, PAS staining was shown to outperform fungal culture and KOH examination [19]. However, calculations frequently do not mention if KOH used staining, which may improve detection over standard KOH. Calculations also frequently use these methods as both ‘reference standard’ and ‘test under evaluation’ in the same calculations which is not an acceptable methodology [20]. Any calculation of diagnostic efficacy must consider an appropriate measure of disease/non-disease outside of the test parameters being evaluated. Furthermore, sensitivity and specificity alone do not provide a full description of diagnostic utility. Positive predictive value and negative predictive value are more thorough parameters for diagnostic assessment but are not typically reported, nor are likelihood ratio, test accuracy, diagnostic OR and Youden Index (sensitivity + specificity—1; “probability of an informed decision”) [19]. Mathematical comparisons of diagnostic methods remain an issue for onychomycosis and should be used with caution when making diagnostic decisions.

## 4. Visual Methods of Diagnosis

Newer visualization methods focus on non-invasive, in-office assessments which can provide faster or real-time information to physicians or staff experienced in these techniques. Non-invasive detection of nail dystrophies and fungal elements suggestive of onychomycosis can quickly focus efforts on patients most likely to need further diagnostic procedures for fungal identification and treatment versus pursuit of differential diagnoses. They may also provide a quick estimation of infection presence or absence post-treatment to aid clinical decisions regarding the need for ongoing therapy. However, these methods do not provide organism identification which may limit their use for determining an adequate treatment plan. 

### 4.1. Dermoscopy

Dermoscopy, also called onychoscopy, provides non-invasive observation of the epidermis, dermal papilla, and deep dermis by eliminating reflected light through polarized photography [21,22]. Characteristic dermoscopic features of onychomycosis include spikes, longitudinal striae, subungual hyperkeratosis, and color changes [23] (Figure 1D). Handheld dermoscopy visualization is fast and low-cost [24]. It may provide a simple method for observing the resolution of nail plate dystrophy following onychomycosis therapy. The limited ability of dermoscopy to identify fungal structures requires combination with another diagnostic method to confirm organism presence/identification [25]. 

Nada and colleagues evaluated the sensitivity of onychomycosis dermoscopy signs as follows: nail spikes—75%; longitudinal striations—82.5%; color changes—95% [16]. Both nail spikes and longitudinal striations led to a 100% specificity rate, but color changes only resulted in a 75% specificity rate. The best diagnostic dermoscopic sign was longitudinal striations [16]. Overall, there is a high level of agreement between dermoscopy signs, clinical and KOH examination [26] and fungal culture [27].

### 4.2. Optical Coherence Tomography

Optical coherence tomography (OCT) uses a non-invasive hand-held device to provide real-time imaging of living tissue on the micron scale. OCT scans perpendicularly through the nail plate, bed and matrix to discriminate minute alterations in these structures that may correlate with fungal infections [28] (Figure 1E). Unaffected nails appear as a band-like layered structure (with some individual differences) in OCT [29].

OCT scans of nails with onychomycosis indicate that irregular surface was the most frequently reported feature and hyperreflective lines were the second most frequent (80.9–83.4% and 71.4–83.4%, respectively) [30]. Dark bands (52.4–66.7%), disturbed architecture (42.9–45.8%), and hyperreflective dots (23.8–50.0%) were also present. Dermatophytomas may show under OCT as an avascular homogeneous mass with a hyperreflective jagged border within an inhomogeneous nail plate [31]. Hyperreflective lines and dots are suspected to be areas of fungal presence and could be a guide for targeting fungal sampling [30]. OCT may also be useful as a quick in-office test for the presence/absence of hyphae post-treatment. Negative controls with non-onychomycotic nail dystrophies are currently lacking and further work is needed to consolidate these findings. When compared to other diagnostic methods, OCT was found to have higher sensitivity but lower specificity than culture, histopathology and KOH [3]. 

### 4.3. Confocal Microscopy

Confocal laser scanning microscopy (CLSM) produces high-resolution imaging by capturing reflection (reflectance confocal microscopy, RCM) or immunofluorescence following laser illumination of the tissue. Illumination of specific wavelength is focused on a single plane in the sample being reviewed, rather than transmitted through the whole sample as in standard light microscopy, and reflected light or fluorescence is directed through a ‘pinhole’ to prevent transmission of out-of-focus fluorescent light when generating images. Multiple images can be generated at a variety of depths to provide ‘optical sectioning’ for review. The technology can provide sharper resolutions and more-detailed structure of objects versus standard light microscopy [32]. It is non-invasive [3,33] and can examine a nail directly in the office in real-time (5–10 min), making it a quick evaluation method [10,34,35]. The technique does have a limited depth of penetration, varying based on scanning laser wavelength, and only moderate levels of sensitivity and specificity have been reported [10,36,37]. Confocal microscopy can provide direct imaging of fungal structures suggestive of onychomycosis, showing as bright filamentous septate hyphae [23,33] (Figure 1F). Devices report use of RCM (830-nm), infrared CLSM (1064-nm) and dual wavelength RCM/CLSM (488-nm and 785-nm) for onychomycosis visualization [3,33,38]. Good sensitivity was noted versus KOH, PAS and culture, but specificity has been lower than PAS and culture [3,38]. 

### 4.4. Ultraviolet Fluorescence Excitation Imaging

Preliminary investigation of ultraviolet fluorescence excitation imaging (u-FEI) detected significant difference in autofluorescent signal intensity of healthy and mycotic nail samples [39]. Average and maximum fluorescence values were 10.2% and 23.9% higher in onychomycosis versus healthy nails. This hand-held imaging device can be used in the clinic for point-of-care non-invasive nail evaluation. This technology, in the form of the Woods Lamp, has a history of use in tinea capitis for identification of *Microsporum* species. This technology spotlights areas of infection which could improve sampling and may also be useful as a fast in-office indicator of fungal reduction/elimination post-therapy. More work is needed to evaluate the use of this technology for onychomycosis versus other nail dystrophies. 

## 5. Organism Identification Methods

Response to antifungal therapy varies with fungal genus/species and reports of dermatophyte resistance to oral therapy are increasing. Non-dermatophyte species are also associated with poorer oral treatment outcomes. Identifying organisms to genus is crucial for proper choice of therapy and evaluation of therapeutic outcomes. The high false negative rate of culture and dermatophyte-focused culture media has prevented development of a complete outline of the dysbiosis associated with onychomycosis. Faster, broader and more reliable organism identification is required and may be aided by these newer methods. These methods do not always indicate fungal viability but provide a counterpoint to the high false-negative culture rate by giving faster and more reliable organism identification, both pre- and post-treatment, for use in making optimum treatment decisions. 

### 5.1. Lateral-Flow-Based Techniques

Lateral flow immunoassays are used in a wide variety of in-office testing, such as urine pregnancy tests, and most recently for COVID-19 antibody rapid testing. These tests provide a clear visual indicator of positive tests represented by a line appearing in the test device after capillary action exposes the prepared specimen to the test area (Figure 2A). The ‘positive test’ line appears following recognition of the test target by a complementary molecule coating the test strip in a horizontal line which is capable of binding to the test target, triggering a visible color change. The low degree of operator skill required makes this type of testing easy and appealing in the office setting, with rapid results provided for evaluation. Lateral flow tests have been developed for assessment of dermatophyte infections, including onychomycosis, using antibodies with high specificity to dermatophyte molecules (e.g., Dermatophyte Test Strip, DermaQuick^®^) [40,41]. Testing can be performed directly in-office with both nail and skin samples. Currently, the presence of dermatophytes can be detected with good specificity using these existing flow devices, but results are not species-specific; cross-reaction has also been noted to a low degree with some non-dermatophyte species [40,41,42,43,44]. These tests are useful to provide indication of a likely dermatophyte infection, but further identification is required to select an optimum therapy. Clinicians need to be aware of which dermatophytes can be detected and which non-dermatophytes are known to cross-react with the device. 

Other molecular targets being investigated include fungal α-1,6 mannan and secreted subtilisin-like protease 6 (Sub6), which could be possible targets for dermatophyte flow assays [45,46]. As fungal genetic knowledge progresses, development of more species-specific assays is feasible and would be a great asset for improving in-office diagnosis. If a species-specific lateral-flow device could be made available, the low-tech in-office convenience would make it a highly valued diagnostic tool.

### 5.2. Polymerase Chain Reaction

It is difficult to summarize polymerase chain reaction (PCR) methods, as nearly every step can be performed in unique ways based on laboratory experience, equipment, preferences and outcome goals. In general, PCR uses fungal-specific molecular primers to amplify fungal DNA in test samples to amounts allowing identification of the fungal organism subtype [47]. Primers can utilize a variety of targets, such as the 28S rRNA, the internal transcribed spacer region of ribosomal DNA, the chitin synthase I gene, or the topoisomerase II gene [48,49]. PCR techniques may also vary in DNA extraction and in PCR product analysis [49]. Conventional PCR generally uses one or several post-PCR steps to produce identifiable genetic fragments, which increases the amount of manipulation needed for identification and provides increased potential for contamination [49,50]. Identification of amplified products uses techniques such as agarose gel analysis/restriction fragment length polymorphism (RFLP) analysis (Figure 2B), ELISA- or probe-based detection, or direct molecular sequencing of fragments [49,50]. Use of real-time PCR was implemented to provide more automated amplification in a ‘closed tube’ system to minimize contamination risk, while monitoring quantities of amplified DNA as measured through increasing intensity of fluorescent dyes or probes hybridizing to the DNA. [49,51]. Genetic identification of fungi via PCR, regardless of methods used, is much faster than culture (hours/days versus weeks), and the availability of commercialized PCR kits for dermatophytosis/yeasts/non-dermatophytes have improved the standardization of testing, but PCR requires highly skilled laboratory technicians and highly specialized equipment to perform testing. PCR also does not typically confirm the viability of identified organisms.

There is confusion in the medical literature concerning real-time PCR. Firstly, use of ‘RT-PCR’ as an abbreviation for real-time PCR is incorrect, and should be restricted to mean ‘reverse transcriptase’ PCR methods. Real-time PCR is properly called quantitative PCR (qPCR) [51]. Secondly, though real-time PCR can provide quantification of material, it does not necessarily identify viable organisms. Quantification does not discriminate between live or dead cells unless testing is modified to remove extracellular DNA and non-viable cell DNA and prevent its use in the PCR reaction (e.g., use of intercalating fluorescent dyes such as propidium monoazide (PMA) and ethidium monoazide (EMA)) [51]. Where ‘real-time’ PCR is discussed in the medical literature, further review is needed to determine if this method is truly assessing only viable material or may be merely a qPCR of any viable/non-viable material. Where attempts have been made to eliminate non-viable material, this testing is particularly useful as clinicians have more certainty of the identified agent being active within the nail at the time of examination, and remaining active post-treatment. As with culture, viability alone does not prove causality and clinicians must consider the identification in conjunction with the clinical presentation and patient history for the best possible diagnosis. 

Identification is limited to those organisms specific to the primers used; clinicians must be aware of which organisms fall within the PCR testing spectrum and which organisms cannot be ruled out by testing [50]. Primers may also amplify contaminants along with causative organisms. PCR can identify mixed infections of NDMs/dermatophytes, and other fungal combinations. Identification of dermatophytes and non-dermatophytes by PCR has been found to be higher than with fungal culture [47,52]. When compared to other diagnostic methods, PCR has shown high sensitivity and good specificity versus KOH and culture, particularly with dermatophyte identification [6,53,54,55,56,57].

### 5.3. Matrix-Assisted Laser Desorption Ionization–Time of Flight (MALDI-TOF)

Matrix-assisted laser desorption ionization–time of flight (MALDI-TOF) is a type of mass spectrometry (MS) that can be used for the analysis of organisms. Test samples are ionized within the spectroscopy instrumentation and accelerated by a known energy source to a detector which provides measurements of the mass-to-charge intensity spectrum of the accelerated particles (Figure 2C). This spectrum represents a ‘molecular fingerprint’ which can be compared to a standardized ‘library’ of known organisms. It has been shown to be a useful tool for the quick and successful identification of dermatophytes [58,59,60]. MALDI-TOF was able to correctly identify many species known to be difficult to identify in the laboratory [60] and can also be used for the identification of clinical yeast including *Candida* spp. and other NDMs [61,62]. 

The MALDI-TOF MS devices are highly-technical laser mechanisms and require complex software for result analysis. Within a laboratory setting performing routine mycology analysis, MALDI-TOF MS can be cost-effective [63]; however, given the need for the equipment and trained staff, this method is not feasible for the average clinician. Laboratories may also need to build their own libraries for MALDI-TOF MS if standard data libraries do not suit their diagnostic needs [6]. Technicians must obtain and maintain cultures for assessment with the device at this time; although testing is technically fast, time for adequate growth of a culture for ID must be factored in to the timing of the final result.

Though it has apparent good sensitivity for identification, utility for diagnosis is limited by the need for a viable pure culture for identification; where a fungal culture is negative, no identification can be performed with this method, in contrast to PCR which can test directly from a clinical sample. More detailed testing of the ability to detect mixed infections is needed. If future use can adapt to the direct use of a clinical sample, utility would be greatly improved. Clinicians need to be aware of which organisms are targeted/not targeted in the ‘library’ for reporting.

### 5.4. Raman Spectroscopy

Raman spectroscopy is a non-destructive chemical analysis based on the phenomenon of Raman scattering, where molecules produce scattering of light (photons) upon vibrational excitation of the molecular bonds [64,65,66]. Upon vibration, most scattering occurs at the same wavelength as the incident light, but some scattering may occur at other wavelengths, known as the Raman scatter, and recorded as the ‘shift’ from the excitation wavelength [64]. Molecules show unique patterns of Raman scatter, captured by Raman spectroscopy as a ‘chemical fingerprint’ [64,67] (Figure 2D). Raman spectroscopy systems typically consist of a laser-generated light source for excitation, with a microscope and detectors to capture the spectra for analysis. The equipment is highly-specialized and complex software is required for the Raman spectral output analysis. Spectra may vary based on sample pre-analytic processing, type of laser used for excitation, use of signal enhancement methods, and analytic data processing techniques [68]. It is generally a quick and low-cost option, [68] but requires an experienced user with good knowledge of Raman technology to set up testing and generate/evaluate the outcomes.

Preliminary testing indicates some success in identifying dermatophytes directly from onychomycotic nails through Raman spectroscopy, as well as *Candida* species and *Scopulariopsis brevicaulis* [69,70]. Surface-enhanced Raman spectroscopy (SERS) uses a nanostructured metal surface to adsorb molecules and enhance the Raman signals and has differentiated several *Trichophyton* species directly from skin samples [71,72].

The ability to detect species-specific infection directly from samples would make Raman spectroscopy a powerful tool for diagnosis of onychomycosis, avoiding culture delays. More work is needed to verify its utility in wider species and specimens; it would be particularly important to verify if mixed species infection can be detected clearly. This method may also have the potential for determining viability of fungi and for in vivo nail scanning in-office, but these remain future goals [69]. As with MALDI-TOF and PCR, identification depends upon the ‘library’ of chemical fingerprints specific to the Raman spectroscopy set-up and sample processing. Clinicians will need to be aware of which species can or cannot be identified by the device library. 

## 6. Artificial Intelligence

In this technological age, artificial intelligence (AI) has demonstrated utility in the diagnosis of cancer, acne vulgaris [73], and psoriasis [74]. The visual assessments required in dermatology may provide large image databases that can be readily used for development of AI diagnosis systems [36]. Currently, AI is not itself a diagnostic ‘method’ for onychomycosis, but instead provides a method to augment visual identification by minimizing the subjectivity of visual assessments. AI assessment algorithms are developed during computer ‘training’, where visual images of known ‘infection’ or ‘non-infection’ status are analyzed, and validation of the resulting data is used to further refine the algorithms with ongoing analysis. Upon ‘training’ completion, unknown images can be assessed as to the likelihood that the image represents fungal structures, or fungal toenail.

AI training develops convolutional neural networks (CNN, also known as ‘deep’ neural networks) for diagnosis of onychomycosis. CNN have successfully differentiated onychomycosis from other nail disorders (e.g., nail dystrophy, onycholysis, melanonychia, subungual hemorrhage, paronychia, subungual fibroma, ingrown nail, pincer nail, and periungual warts) using photographic assessment, with the CNN outperforming most of the 42 dermatology reviewers [75]. Another AI review of toenail photographs provided comparable onychomycosis diagnosis versus dermatologist photo evaluation and toenail dermoscopy [76]. For histopathology slides of PAS-stained nail clippings, diagnosis by a CNN was comparable to dermatopathologists, with only the most experienced dermatopathologists exceeding diagnosis rates of the CNN [77]. Assessment of greyscale KOH slide images for fungal structure presence by AI models showed 95.90–95.98% accuracy versus a clinician average accuracy of 72.8% [78].

AI provides a method of automating routine visual assessments which could reduce the need for highly skilled pathologists and reduce the time needed for assessment completion. However, development of AI systems requires highly skilled programmers and high numbers of reliable quality images for computer training. Availability of the current AI techniques may be useful for clinicians who lack experience in onychomycosis assessment. For potential patients, access to online AI photo assessment [75] may prompt a visit to a physician sooner rather than later, with earlier treatment being associated with increased treatment success. AI technology in onychomycosis is only in the early stages of development; it remains to be seen how far this technology can be taken for onychomycosis diagnosis.

## 7. Conclusions

In addition to the established methods for diagnosing onychomycosis, of which there are many, there are more recent and currently emerging technologies. Many new technologies require further research and consideration before their use in clinical practice can become mainstream. Moreover, the efficacy of such methods can vary based on many factors (e.g., skill of the technician or physician, variance in techniques). Obtaining the complete and correct identity of all infective fungal organisms, particularly where non-dermatophyte molds are present, is needed to ensure appropriate treatment. Sensitivity and specificity measures are not necessarily the best method of assessment due to problematic reference standards, and may be inappropriately or incompletely reported in study outcomes. Overall, most of these diagnostic tools have advantages that offer useful assistance to the diagnoses of onychomycosis but none can currently replace the established methods entirely. However, given the uncertainty of the established methods, a combination of new and/or conventional methods is likely to be prudent for the most thorough diagnosis and to ensure optimum antifungal therapy is selected for the infecting organism.

Onychomycosis typically shows relatively slow progression. Physicians have time to perform a thorough diagnosis without compromising patient outcomes, even with the slowest diagnostic methods. However, patients may be frustrated with waiting for a diagnosis or needing multiple diagnostic visits. The ideal diagnostic techniques need to work for both the clinician and the patient, in terms of timelines, costs and reporting outcomes. Clinicians will need to ‘look beyond the numbers’ when determining which techniques will provide the best options for the patient’s overall assessment and treatment. Regardless of which methods may become reference standards in the future, the continued increase in diagnostic options will increase onychomycosis knowledge and assist clinicians in providing optimal onychomycosis care.

## Figures and Tables

**Figure 1 jof-08-00464-f001:**
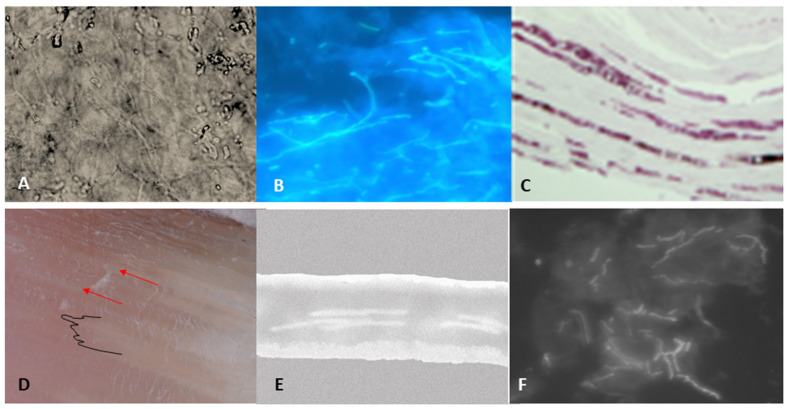
**Examples of outputs produced for visual methods of identification**. (**A**) Standard KOH exam photo showing hyphae; (**B**) KOH exam of hyphae using fluorescent microscopy; (**C**) Septate black hyphae in the nail plate (periodic acid-Schiff staining ×400) from Figure 2b in [7]; (**D**) Illustration of dermoscopy onychomycosis signs in the nail plate: jagged proximal edge (black outline), longitudinal striae (red arrows); (**E**) Optical coherence tomography: digital illustration of output through nail plate showing hyphae; (**F**) Confocal microscopy: illustration of visualized hyphae.

**Figure 2 jof-08-00464-f002:**
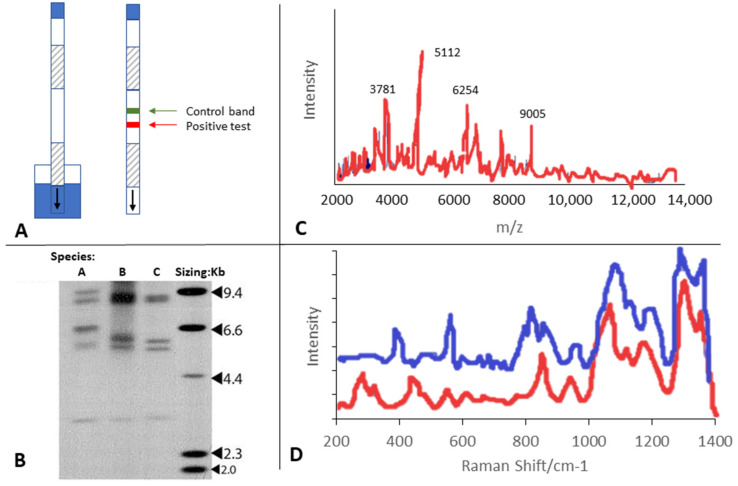
**Examples of outputs produced for organism/molecular identification**. (**A**) Illustration of Lateral Flow tests trips; (**B**) Example of RFLP output after PCR processing showing 3 samples being processed; (**C**) Digital illustration of MALDI-TOF spectrum output; (**D**) Digital illustration of Raman spectrum output (species 1—red; species 2—blue).

**Table 1 jof-08-00464-t001:** Summary of Diagnostic Methods for Onychomycosis.

Diagnostic Method	Available for Use?	Viability Indicated?	ID Outcome	Relative Use Details
**Visual Identification**				
KOH preparation	Yes	No	Fungal presence/absence via yeast and hyphae	Tech costs: Low Performance: Lab tech/physician; in office or lab Time requirements: <1 h
PAS staining	Yes	No	Fungal presence/absence via yeast and hyphae	Tech costs: Moderate—High Performance: Lab tech; in lab Time requirements: <1 h—hours
Dermoscopy	Yes	No	Nail infection vs. other abnormalities	Tech costs: Low Performance: Physician; in office Time requirements: <1 h
Ultraviolet fluorescence excitation imaging (u-FEI)	In development	No	Fungal presence/absence	Tech costs: Low Performance: Physician; in office Time requirements: <1 h
Confocal microscopy	Yes	No	Fungal presence/absence via yeast and hyphae	Tech costs: High Performance: Lab tech; in lab Time requirements: <1 h—hours
Optical coherence tomography	Yes	No	Fungal presence/absence via yeast and hyphae	Tech costs: High Performance: Lab tech; in lab Time requirements: <1 h—hours
**Organism Identification**				
Fungal culture	Yes	Yes	Dermatophytes/ NDMs/yeasts	Tech costs: Low Performance: Lab tech/experienced mycologist; in lab Time requirements: Weeks
PCR	Yes	No—(specialized qPCR only)	Primer-dependent: Dermatophytes/ NDMs/yeasts	Tech costs: High Performance: Lab tech/specialist; in lab Time requirements: Hours-days (wide variation depending on methods used)
MALDI-TOF mass spectroscopy	Yes	Yes (positive culture as test sample)	Library-dependent: Dermatophytes/ NDMs/yeasts	Tech costs: High Performance: Lab tech/specialist; in lab Time requirements: <1 h—hours for test (Weeks for culture preparation)
Raman spectroscopy	In development	No	Library-dependent: Dermatophytes/ NDMs/yeasts	Tech costs: High Performance: Lab tech/specialist; in lab Time requirements: <1 h—hours
**Visual Augmentation**				
Artificial Intelligence (AI)	In development	No	Technique-dependent: Fungal presence/absence OR Nail infection vs. other abnormalities	Tech costs: Moderate Performance: Specialized software; computer device or online Time requirements: Minutes—hours

**Acronyms:** potassium hydroxide (KOH), periodic acid-Schiff (PAS), polymerase chain reaction (PCR), real time/quantitative polymerase chain reaction (qPCR), matrix-assisted laser desorption ionization–time of flight (MALDI-TOF).

## Data Availability

Not applicable.

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
