# Peer review of "Diagnosing Onychomycosis: What’s New?"

_jof, 2022, doi:10.3390/jof8050464_

Round 1

Reviewer 1 Report

Comments to the Author

RE: Diagnosing Onychomycosis: What’s New? (jof-1702203).

“The manuscript entitled Diagnosing Onychomycosis: What’s New?" reviewed properly the established methods of diagnosing onychomycosis and other diagnostic methods under development. The manuscript has appropriately been written and has good structure; however, some points must be corrected to promote the quality of this review article.

Abstract:

  • It is essential to briefly mention the new methods and devices which has been introduced in manuscript.

Methods:

  • Line 37-38 (The dermatophyte Trichophyton rubrum is the most common organism causing infection). This sentence should be replaced by "dermatophytes Trichophyton rubrum and interdigital are the most common organisms causing infections" according to following paper and book.
    1-Onychomycosis in the 21st Century: An Update on Diagnosis, Epidemiology, and Treatment
    2-book: dermatophye and dermatophytosis
  • Page 2: Table 1, PCR; One of the main advantages of PCR test is speed.
    why mentioning days? For diagnosis, 2 hours extraction of DNA and 3-4 hours PCR with specific primers leads to reliable results.
  • Page 4: Fungal culture, this manuscript mainly focused on diagnosis of onychomycosis and not only Tinea unguium. So SCC media is suitable for dermatophytes culture and not for
  • Page 5: Dermascopy: Could you please add a figure for dermoscopy if available, more useful in diagnosis.
  • Page 7: Figure 2: Part B, RLFP, this part is not clear. Perhaps more clear RFLP can be adopted from the following reference (Use of single enzyme PCR restriction digestion barcode targeting the internal transcribed spacers (ITS eDNA) to identify dermatophyte species).

Author Response

Thank you for your helpful comments. The following changes have been made:

  1. Abstract: new methods have now been listed here as requested (lines 21,23-24)
  2. T. interdigitale has been added to introduction as a relevant organism (line 40)
  3. Some methods of PCR do take more than several hours, and require days - a statement has been added to the table to clarify that times are method-dependent. (Table 1 - PCR final column)
  4. We are not sure what the comment regarding media was asking. We are not intending to be recommend certain media - choice can be made based on lab goals, and we only wanted to indicate that media can be altered as needed.
  5. Per your suggestion, we have updated Figure 1 to include a diagram illustrating general findings in dermoscopy of onychomycosis (Figure 1D)
  6. Figure 2 now has an updated illustration for RFLP (Figure 2B). We do not have access to a specific dermatophyte RFLP slide. Also, as RFLP techniques vary, there is no standard illustration of what to expect. This is a general RFLP output only, to show how different species appear on the RFLP output generally.

Reviewer 2 Report

Comments:

Page7, Line 226

There is several PCR methods for fungal identification, but you just talk about RFLP and qPCR. Why?

Page 7, Line 233, Please add refrence for RFLP techniques.

Page 7, Line 235, Change Figure 2A to Figure 2B

Page 7, Line 272, Change Figure 2B to Figure 2C

Page 9, Line 300, Change Figure 2C to Figure 2D

Page 9, Line 333, what is CNN?

Author Response

Thank you for your helpful comments. Please see our list of changes below:

  1. re: line 226 comments- We have rewritten the PCR section to more clearly indicate that there are a variety of methods at all stages of PCR. (lines 230-250)
  2. re: line 233 comment - As we have minimized RFLP reference now, we have not added specific references for this method of PCR.
  3. re: lines 235, 272, and 300 comments - We have now corrected the in-text references to Figures 2A,B,C and D (lines 208, 241, 281, and 309) 
  4. re: line 333 comment - we have updated the sentence with CNN abbreviation, to make the definition more clear (it is Convolutional Neural Network) - line 342

This manuscript is a resubmission of an earlier submission. The following is a list of the peer review reports and author responses from that submission.